**Data Availability Statement:** All relevant data are within the manuscript and its Supporting Information files.

**Funding:** RGW; PJA Grant #NA140AR0110267, NOAA Ocean Exploration and Research https://oceanexplorer.noaa.gov The funders had no role in

# Deep coral habitats of Glacier Bay National Park and Preserve, Alaska

**Élise C. Hartill**[1]*, **Rhian G. Waller**[1,2], **Peter J. Auster**[3,4]

**1** Darling Marine Center, School of Marine Sciences, University of Maine, Walpole, Maine United States of America, **2** Sven Lovén Centre, Tjärnö, University of Gothenburg, Strömstad, Sweden, **3** Department of Marine Sciences, University of Connecticut, Groton, Connecticut, United States of America, **4** Mystic Aquarium–Sea Research Foundation, Mystic, Connecticut, United States of America

* elise.hartill@maine.edu

## Abstract

Glacier Bay National Park and Preserve (GBNPP) in Southeast Alaska is a system of glaciated fjords with a unique and recent history of deglaciation. As such, it can serve as a natural laboratory for studying patterns of distribution in marine communities with proximity to glacial influence. In order to examine the changes in fjord-based coral communities, underwater photo-quadrats were collected during multipurpose dives with a remotely operated vehicle (ROV) in March of 2016. Ten sites were chosen to represent the geochronological and oceanographic gradients present in GBNPP. Each site was surveyed vertically between 100 and 420 meters depth and photo-quadrats were extracted from the video strip transects for analysis. The ROV was equipped with onboard CTD which recorded environmental data (temperature and salinity), in order to confirm the uniformity of these characteristics at depth across the fjords. The percent cover and diversity of species were lowest near the glaciated heads of the fjords and highest in the Central Channel and at the mouths of the fjords. Diversity is highest where characteristics such as low sedimentation and increased tidal currents are predominant. The diverse communities at the mouths of the fjords and in the Central Channel were dominated by large colonies of the Red Tree Coral, *Primnoa pacifica*, as well as sponges, brachiopods, multiple species of cnidarians, echinoderms, molluscs and arthropods. The communities at the heads of the fjords were heavily dominated by pioneering species such as brachiopoda, hydrozoan turf, the encrusting stoloniferan coral *Sarcodyction incrustans*, and smaller colonies of *P. pacifica*. This research documents a gradient of species dominance from the Central Channel to the heads of the glaciated fjords, which is hypothesized to be driven by a combination of physical and biological factors such as glacial sedimentation, nutrient availability, larval dispersal, and competition.

## 1. Introduction

While patterns of species diversity and ecosystem processes are relatively well studied in shallow coral reef ecosystems [1–3], there are fewer detailed studies of cold-water coral ecosystems [4–7]. Yet these ecosystems play a similar ecological role as their shallow counterparts by

study design, data collection and analysis, decision to publish, or preparation of the manuscript.

**Competing interests:** The authors have declared that no competing interests exist.

serving as the foundation for, and facilitating ecological processes that sustain, high biodiversity communities [8–11] and thus providing vital ecosystem services [12].

## 1.1 Fjord ecosystems

Fjord systems present a unique opportunity to study deep-sea organisms at shallower depths due to deep-water emergence. Deep-water emergence is a phenomenon where usually deep-sea species live at shallower depths than usual in high latitude fjord ecosystems where the oceanographic characteristics mimic a deep-sea environment [13, 14]. Glacial meltwater, and, in some cases, high precipitation levels, form a coherent layer of freshwater that sits on top of higher density ocean water. Brackish coastal water attenuates light more than oceanic water due to higher concentrations of particulate matter, including colored dissolved organic matter (CDOM) from glacial meltwater [15]. In some fjords, subglacial freshwater discharge can occur as well, causing upwelling and vertical mixing thereby increasing turbidity, and darkness, at depth [16, 17]. This reduced light is increased further by shading from steep fjord walls and narrow deep basins. The water temperatures in the fjords are similar to those found at bathyal depths and the complex bathymetry allows for strong tidal currents and circulation of well oxygenated, nutrient-rich water [13, 14, 18]. In areas such as Alaska, Chile, New Zealand, and Scandinavia, cold-water corals grow on the steeply sloping rock walls at depths as shallow as five meters [13, 14, 19, 20]. Due to the glacial sedimentation typical of sub-polar fjords, faunal diversity and biomass generally declines from the outer to inner fjords [21, 22]. This general pattern of fjordic diversity cline has been observed in a number of fjords located in different geographical locations including Norway, Greenland, the Canadian Arctic, Scotland, and New Zealand [22]. The diversity clines in fjords are attributed to a number of factors including environmental disturbance unique to fjords (e.g. glacial activity), colonization barriers resulting from geomorphological features (e.g. sills) or distance from species pool [22, 23]. Studies are finding that contrary to earlier assumptions, fjord fauna are not only a subset of offshore species pools, but that there are also locally occurring fjordic species that contribute to species richness [22, 24]. Environmental factors such as substratum type, water temperature, depth, and benthic food supply are important determinants of community structure [21, 25]. A study of two Svalbard fjords and the adjacent continental shelf showed that bottom water temperature, an indicator of Atlantic or Arctic water mass influence, explained over a third of the variability in functional trait diversity (i.e. predators, mobile scavengers, sessile suspension feeders, and detritivores) [21]. Warming due to climate change is likely to increase glacial melt, calving and sedimentation, which could potentially decrease megafaunal biomass and functional diversity in fjord environments, leading to a shift towards suspension-feeding and detritivore communities [21].

## 1.2 Glacier Bay National Park and Preserve

Cold-water coral ecosystems have been observed in Glacier Bay National Park and Preserve (GBNPP) in Southeast Alaska. Biologists confirmed the presence of the cold-water coral *Primnoa pacifica*, commonly known as the Red Tree Coral, during shallow water SCUBA surveys in 2003 [14]. In 2010, sixteen sites were surveyed using a monochrome video pencil-camera across the Central Channel, the East Arm, and the West Arm of Glacier Bay at depths between 20 and 180 meters. That survey confirmed the extensive presence of cold-water coral habitats in the deeper areas of GBNPP (R Waller, pers comm.).

Glacier Bay National Park and Preserve encompasses a Y-shaped, glacially formed system of fjords located northwest of Juneau, in Southeast Alaska (northeast Pacific Ocean; Fig 1). It is bounded by the Fairweather Range to the west, the Chilkat Range to the east, and the Saint

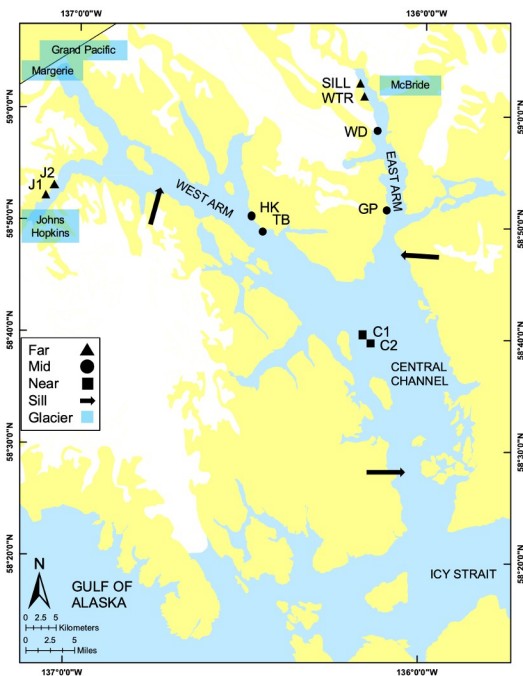

**Fig 1. Map of studies sites in Glacier Bay National Park & Preserve.** The ten study sites are marked: far sites (>50km from tidewater glaciers), mid sites (20-40km from tidewater glaciers), and near sites (<10km from tidewater glaciers). Sills are indicated with arrows, tidewater glaciers are represented by blue boxes.

Elias and Takhinsha Mountains to the north. Glacier Bay contains the largest active glacier complex in the world outside of Antarctica and Greenland, and has a complex ice history; its basin morphology is the result of several glacial events. At the maximum of the Little Ice Age (LIA) about 250 years ago, the area was covered with an extensive icefield [26]. Glacier Bay was historically inhabited by the Huna Tlingit, who have moved in and out of the Bay for centuries as the glaciers advanced and retreated [27]. In 1925, Glacier Bay was designated a National Monument, and in 1980, was re-designated as GBNPP with the signing of the Alaska National Interest Lands Conservation Act [28]. In 1999, federal legislation was passed to exclude commercial fishing and to allow for subsistence fishing in GBNPP, marking the culmination of decades of efforts by stakeholders [29]. The National Monument was founded with the principle mandate of "preserving the opportunity to conduct scientific studies," and thus remains a relatively pristine environment. Although GBNPP is presently free of commercial fishing activities, it is not exempt from the major anthropological disturbances that stem from climate change.

Glacier Bay encompasses the East Arm fjord, West Arm fjord, and Central Channel, comprising an area of 1,255 km$^2$ and a total length of 105 km. Glacier Bay has been described as a combination of a stratified deep basin estuary and a tidally mixed estuary [30]. There are multiple sills of varying depths in Glacier Bay and its tributary arms, there is a shallow sill (25 meters depth) at the entrance of Glacier Bay and another sill (60 meters depth) at the entrance of the East Arm, as well as a sill (240 meters depth) in the upper section of the West Arm, just southeast of Tarr Inlet [30, 31]. The deep basins located behind the sills reach depths of up to 450 meters in the West Arm's central basin, and 300 meters in the East Arm's central basin as well as in the Central Channel. The East Arm, also known as Muir Inlet, encompasses Adams Inlet, which branches off to the east, and Wachusett Inlet to the west. The East Arm has one

tidewater glacier, McBride Glacier, which has been retreating since the 1960s. There are several grounded glaciers in the East Arm, including Riggs Glacier, which grounded in the mid-1980s, and Muir Glacier, which had extreme rates of retreat and calving beginning in the 1890s and grounded at the head of the fjord in 1993 [32]. The West Arm terminates in Tarr Inlet to the northwest, with Johns Hopkins Inlet branching off just southwest of Tarr Inlet. Rendu Inlet and Queen Inlet branch off the east side of the West Arm. Johns Hopkins Glacier and Gilman Glacier are two tidewater glaciers in Johns Hopkins Inlet that are currently advancing. Margerie Glacier is a hanging glacier in Tarr Inlet; its terminus was relatively stable until it resumed retreating in recent years. These two fjords, the East Arm and West Arm, are joined together in a central channel that leads over the submerged terminal moraine and out into the Southeastern Alaskan Continental Shelf and the Pacific Ocean via the Icy Strait.

Glacier Bay experiences a wet and moderate maritime climate, and freshwater runoff from precipitation is naturally heightened by the steep sloping walls of the fjords [33]. Although many of the glaciers in GBNPP are now grounded, they contribute a consequential amount of glacial-melt water and fine sediment [34, 35]. At the lower latitudes of the fjord, where deglaciation took place decades or centuries ago, the long-established terrestrial vegetation contributes to a more diverse and abundant underwater benthic community by reducing runoff and sediment erosion that facilitates settlement and survivorship of suspension feeding species [36]. In Glacier Bay and many glaciated fjord estuaries, freshwater input from glacial melt and stored and direct precipitation appears to be the greatest driver of oceanographic properties. This freshwater input affects water column stratification and flow dynamics and introduces suspended and dissolved materials [31]. Stratification varies seasonally, with the greatest stratification occurring in the summer and fall months and increasing in strength with distance from the mouth of the bay [31]. Strong stratification leads to heightened light attenuation at the heads of the fjords thereby affecting the biological activity in those areas. The relative influence of tidal currents in Glacier Bay should also be noted, the tidal currents are high in the lower bay and lower in the rest of the bay and especially in the upper reaches of the East and West Arms [31]. The surface water turbidity is highest at the heads of the fjords but is highly variable both spatially and temporally throughout the year [31]. The surface waters of the East Arm have consistently lower salinity and higher stratification than those in the West Arm due to the differences in the rates of freshwater discharge for each tributary and in the circulation patterns, a result of basin topography and the 60 meter deep sill at the entrance of the East Arm [31]. The central basins of Glacier Bay, where there is decreased sedimentation, higher light levels, intermediate stratification and upwelling of nutrient-rich oceanic water, that result in higher sustained concentrations of chlorophyll α, may have optimal conditions for aggregations of benthic suspension-feeding organisms [31]. This is also where deep-water emergence of *P. pacifica* has been well documented. *Primnoa pacifica* colonies observed at the mouths of the fjords are more robust than those found at the heads of fjords, where glacial influence is increased [14]. Carney et al. in 1999 found that shallow benthic species composition differed greatly between glaciated fjords in GBNPP and non-glaciated fjords in Southeast Alaska [36]. They also found that shallow benthic species richness and abundance increased significantly from the head of the glaciated fjords to the mouths of those same fjords, citing glacial influence as a primary driver of the observed differences.

## 1.3 Primnoa pacifica

*Primnoa pacifica* (Kinoshita, 1907) is an alcyonacean in the family Primnoidae found only in the North Pacific Ocean [37]. Mature *P. pacifica* colonies are massive tree or bush-like structures, often exceeding two meters in height and several meters in width [8] with large

individuals exceeding 100 years of age [38]. Their depth of occurrence ranges between 6 and 1029 meters [37, 39], though they are most commonly found at around 500 meters on seamounts and along the continental shelf edge of the Northeast Pacific. The large, complex structure of *P. pacifica* colonies provides habitat for a diverse community of associated species, some of which (such as rockfish and crabs) are economically important in the Gulf of Alaska and Bering Sea [8, 40, 41]. *Primnoa pacifica* colonies have a positive effect on the biodiversity of the community therefore exhibit keystone species characteristics as defined by Power et al. in 1996 [14, 41–43].

## 1.4 Objective

The objective of this study was to examine how glacial distance influences the composition of coral communities in GBNPP Alaska. This study expands on earlier surveys to examine the bathyal benthic community structure between 100 and 420 meters in GBNPP using a remotely operated vehicle (ROV). The age of deglaciation at each of the study sites differs due to the variable rates at which glacial retreat occurred across Glacier Bay after the LIA [14, 26, 32, 35]. The results reported here demonstrate a gradient of diversity that informs our understanding of the biological and ecological processes (succession, competition, etc.), and physical processes (sedimentation, stratification, etc.) that may drive patterns of benthic community composition.

## 2. Methods

### 2.1 Site selection

The ten sites were selected from multibeam bathymetric maps using peer-reviewed knowledge of the habitat characteristics that are associated with *P. pacifica* communities [31, 41, 44, 45]. The sites were chosen to represent the geochronological and oceanographic gradients of Glacier Bay. Underwater video from vertical transects was collected at depths between 100 and 420 meters using a remotely operated vehicle (ROV) in March of 2016 (National Park Service Research Permit GLBA 00653). Due to logistical limitations, only ten sites were chosen to capture the latitudinal gradient of Glacier Bay's fjord system. Four sites were located in the East Arm, four in the West Arm, and two in the Central Channel. The ten sites were further classified into zones corresponding to their proximity to tidewater glaciers and to the terminus of the fjord in which they are located as "near" (<10km), "mid" (20-40km) or "far" (>50km) (Fig 1). Characteristics of each site, length of transect, and area surveyed are summarized in Table 1.

### 2.2 Survey method

The ROV Kraken2 (University of Connecticut) was used to conduct ten multipurpose dives (i.e., visual survey, specimen collection) principally focused on vertical walls in the fjords. For each dive the ROV initially descended to the seafloor near the central axis of the fjord to avoid collisions with precipitous terrain and then was driven towards the base of the wall where the transects began. Transects were generally conducted vertically, ascending the fjord wall towards the surface. Due to the multipurpose nature of the ROV dives, the ascent was non-linear and exploratory, at times traveling horizontally to avoid collision with geologic features or to collect specimens. The dives at the two Central Channel sites were conducted horizontally over a low sloping landscape as opposed to along the walls of the fjords. The ROV was equipped with paired parallel scaling lasers that were set at 10 cm apart for image calibration, as well as a conductivity, temperature and depth sensor system (CTD) Sea-Bird SBE-19 (Sea-

**Table 1. ROV metadata.**

| Site Name & Abbreviation | Location | Distance to Tidewater Glacier (km) | Average Temperature at Depth (˚C) | Temperature SE | Salinity at Depth (psu) | Salinity SE | Dominant Substrate | Dominant Slope (degrees) | Transect Length (m) | Area Surveyed (m²) |
|---|---|---|---|---|---|---|---|---|---|---|
| Johns Hopkins 1 (J1) | West | 4.06 | 6.02 | 9.36 E-04 | 30.08 | 1.56 E-03 | Bedrock/Soft Sed | >30˚ | 98 | 24 |
| Johns Hopkins 2 (J2) | West | 5.54 | 6.02 | 6.05 E-04 | 30.83 | 7.33 E-04 | Bedrock | >30˚ | 221 | 56 |
| White Thunder Ridge Sill (SILL) | East | 6.46 | 6.02 | 2.43 E-04 | 30.69 | 8.45 E-04 | Bedrock | >30˚ | 71 | 43 |
| White Thunder Ridge (WTR) | East | 8.62 | 6.00 | 3.26 E-04 | 30.8 | 1.01 E-03 | Bedrock | >30˚ | 127 | 34 |
| West Dahl Point (WD) | East | 12.45 | 6.02 | 6.05 E-04 | 30.7 | 4.09 E-04 | Bedrock/Silt | >30˚ | 51 | 27 |
| George's Point (GP) | East | 25.26 | 6.01 | 4.99 E-04 | 30.74 | 3.35 E-03 | Bedrock/Silt | >30˚ | 159 | 47 |
| Happy Knobb (HK) | West | 40.09 | 6.17 | 1.44 E-03 | 30.91 | 3.26 E-03 | Bedrock | >30˚ | 147 | 75 |
| Tidal Bulge (TB) | West | 43.92 | 6.09 | 5.09 E-04 | 30.91 | 1.82 E-03 | Bedrock | >30˚ | 82 | 41 |
| Central Channel 1 (C1) | Main Bay | 48.05 | 6.12 | 3.98 E-03 | 30.91 | 1.21 E-01 | Silt | <30˚ | 55 | 24 |
| Central Channel 2 (C2) | Main Bay | 49.55 | 6.12 | 3.98 E-03 | 30.91 | 1.21 E-01 | Bedrock | <30˚ | 78 | 24 |

Bird Electronics Inc. Bellevue, WA). Video was transmitted from the vehicle over a fiber-optic network in 1080i format and recorded in high-definition MP4 files to facilitate replay and analysis.

Video from the transects was non-linear, therefore a series of non-overlapping photo quadrats was extracted in order to control for the area surveyed (Table 1). Frame captures of the video were taken at every location where the ROV paused for a short period to capture a photo quadrat, which, because specimen collection and exploration were also goals of the cruise, coincided with the presence of epibenthic fauna. A 10x10 cm grid was superimposed on each image (using ImageJ Version. 1.51)[46], calibrated with the scaling lasers, which resulted in 1m² quadrats in order to estimate percent cover of taxa. Quadrats were also ground-truthed by examining video footage around the frame capture in order to assure classification of categorical substrate characteristics and presence and percent cover of small and cryptic taxa.

The CTD collected conductivity, temperature, density and salinity data at 2 second intervals. Depth, latitude, and longitude were recorded for the ROV throughout the dive using an ultra-short baseline tracking system for the vehicle and GPS for the ship position. The temperature and salinity data from collection depths were parsed in Microsoft Excel (16.20) and sample statistics were compared across sites. Turbidity was qualitatively assessed as "low" or "high". Dominant substrate texture was visually assessed according to the Wentworth grade classification [47] as follows: bedrock, boulder, cobble, pebble, sand, silt or shell. Slope was classified as low to medium (<30˚) or high (>30˚). These characteristics were scored to assess dominant habitat type at each site and overall heterogeneity between sites (Table 1).

## 2.3 Community assessment

Megafaunal species (i.e. larger than 1 cm) were identified to the lowest possible taxon and verified by experts or ground-truthed using specimen collections. Identification was usually to the

species level but there were a number of organisms that could only be resolved to the class or family taxonomic levels (e.g. demospongiae or sabellidae). The majority of organisms in the transects were quantified using percent cover. However, rare species [48] that comprised less than 5% of the total area surveyed across quadrats were enumerated and appear only in the analysis of species richness. Rare species are important to take into account due to their potential to contribute to community stability by providing functional redundancy [22, 48]. Relative frequency of occurrence for each taxon was calculated by dividing the number of quadrats in which a species was observed (using presence-absence data) by the total number of quadrats at each location.

*Primnoa pacifica* colonies were classified into one of four size categories, using the paired parallel lasers for measurement, as follows: "1" describes individuals with a height of less than 0.25m, "2" refers to individuals with height between 0.25m and 0.50m, "3" refers to those between 0.50m and 1m, and size "4" individuals were larger than 1m in height and width. *Primnoa pacifica* individuals that were smaller than 10cm were also noted, these presumed juvenile *P. pacifica* [13] or "sprigs" were recorded in order to identify areas where *P. pacifica* had recently recruited to the substrate. New recruits and size classes are important to record because they indicate that there are different age cohorts in these populations. This may be an indication that Glacier Bay's population is reproductively successful, whereas researchers believe that *P. pacifica* populations in some of Alaska's other southeastern fjords are not currently successfully reproducing [49]. Portions of transects where *P. pacifica* colonies were present in high abundance and large size that the substrate and other organisms beneath were not visible were classified as areas of dense "thicket habitat" [41] in the data set. Due to the arboreal and variable morphology of *P. pacifica*, determining coral cover or biomass based on video footage was difficult. In this analysis, coral cover was recorded in two-dimensional percent cover (as for other taxa), acknowledging that a large colony produces significant three-dimensional ecological space while its base (i.e. actual area of attachment to the seafloor) occupies a small area of the quadrat.

## 2.4 Statistical analysis

Sample statistics were calculated for the environmental data (temperature and salinity) at sampling depth and compared across sites to assess whether these variables were uniform. The diversity estimates and multivariate comparisons of megafaunal communities in GBNPP described hereafter were conducted using Primer 6 software (PRIMER-E, Ivybridge, UK). Species accumulation curves (*S* observed) and species richness estimates (CHAO 1, 999 permutations) were calculated using species presence-absence data in order to assess richness and whether sampling effort adequately captured the species diversity. Relative frequency of species was calculated in order to visualize and compare community composition at each site. In order to determine the similarities between sites, the percent coverage data were standardized, and squareroot transformed [50] to allow for contributions from rare and common species, then a Bray-Curtis similarity matrix was calculated for non-metric multidimensional scaling (nMDS) and complete-linkage hierarchical cluster analysis. Next, one-way analysis of similarity (ANOSIM) routines with 999 permutations were conducted to determine whether there were statistically significant differences in community assemblages among sites with distance from glaciers as the factor. A two-way nested ANOSIM routine was conducted with distance to glacier (near, mid, far) and location (East, West, or Central Channel) as factors to determine whether there were statistically significant differences in community assemblages based on which fjord the sites were located in. Lastly, a similarities percentage routine (SIMPER) was conducted to identify which species were driving the observed differences between geographical site groupings.

## 3. Results and discussion

### 3.1 Environmental characteristics

The dominant habitat type in the near glacier and mid fjord sites was steeply sloped hard substrate. The sites that had a notable amount of silt and turbidity (visually assessed qualitatively) were J1 and J2, and WDP and GP, where the silt was primarily accumulated on horizontal steps in the walls. The Central Channel sites were shallow sloping or horizontal seabed as opposed to vertical wall transects and were comprised of a mixture of hard substrates, silt, and barnacle reefs (Table 1). The area surveyed at each site ranged from 24 m$^2$ at C1, C2 and J1, to 75 m$^2$ at HK. The large disparity in area surveyed was an effort to collect a comparable amount of data from megafaunal assemblages at each site, i.e. some sites presented nearly continuous dense assemblages of fauna versus other sites where assemblages of megafauna were patchy. The site averages for the environmental data were relatively narrow, between 6.00˚C and 6.17˚C for bottom temperature, and between 30.08 psu and 30.91 psu for salinity (Table 1).

### 3.2 Species diversity

The species accumulation curves (*Sobs*)indicated that the sampling effort adequately captured the species diversity at each study site as each curve was asymptotic or near-asymptotic (Fig 2A). The Chao 1 richness estimator indicated that generally, once 25 quadrats were sampled, there were not substantially more species that were predicted to be found in each zone (Fig 2B, S1 Fig though 3). There were 31 taxa identified from analysis of video records (Table 2) and there were five dominant taxa present in all three zones: *P. pacifica* (Fig 3A), the brachiopod *Laqueus californicus* (Fig 3E), hydrozoan turf (Fig 3D), the encrusting stoloniferan coral *Sarcodyction incrustans* (Fig 3C) and hexactinellid sponges. The majority of sessile taxa observed were cnidarians (e.g. *P. pacifica*, anemones, solitary cup coral *Caryophyllia arnoldi*, and hydrozoan turf) followed by brachiopods and porifera. The most abundant mobile taxa observed were echinoderms, such as brittle stars, basket stars, sea cucumbers, urchins and sea stars.

### 3.3 Patterns of community structure

The near-glacier sites in the East Arm (WTR and SILL) and in the West Arm (J1 and J2) had low average species richness and abundance relative to the rest of the sites (Fig 4). The mid-fjord sites in the East Arm (WD and GP) had higher species richness and abundance. The mid-fjord sites in the West Arm (HK and TB) had the highest abundance of large (>50cm in height/width) and dense *P. pacifica* colonies. Of the 395 quadrats analyzed, only seven quadrats were devoid of *Primnoa pacifica*, two of the quadrats were at SILL and five were located at HK. Sites HK and GP had lower species richness than expected, which we attribute to the difficulty of observing the below-canopy substrate due to the dense coral canopy and not to decreased diversity overall. Indeed, in *P. pacifica* thicket habitats, diversity is expected to increase with the surface area of *P. pacifica* [8, 41].The sites that were located furthest from glaciers, C1 and C2, had the highest species richness, evenness and abundance (Fig 4).

It is also important to note however that the transects at these sites (TB and HK) were conducted at approximately 100 to 200 meters deeper than the rest of the sites (Table 1). Studies have shown that coral colonies are generally larger as depth increases in fjord environments [13]. Even a relatively small depth difference may have an effect on a number of biotic and abiotic factors that organisms experience. Although food availability is usually negatively correlated with depth [25], increased food supply at depth on the local scale can be increased by oceanographic events, local geomorphology, lateral advection, etc. [21, 51]. For example, changes in topography on a slope increases current velocities, potentially increasing the

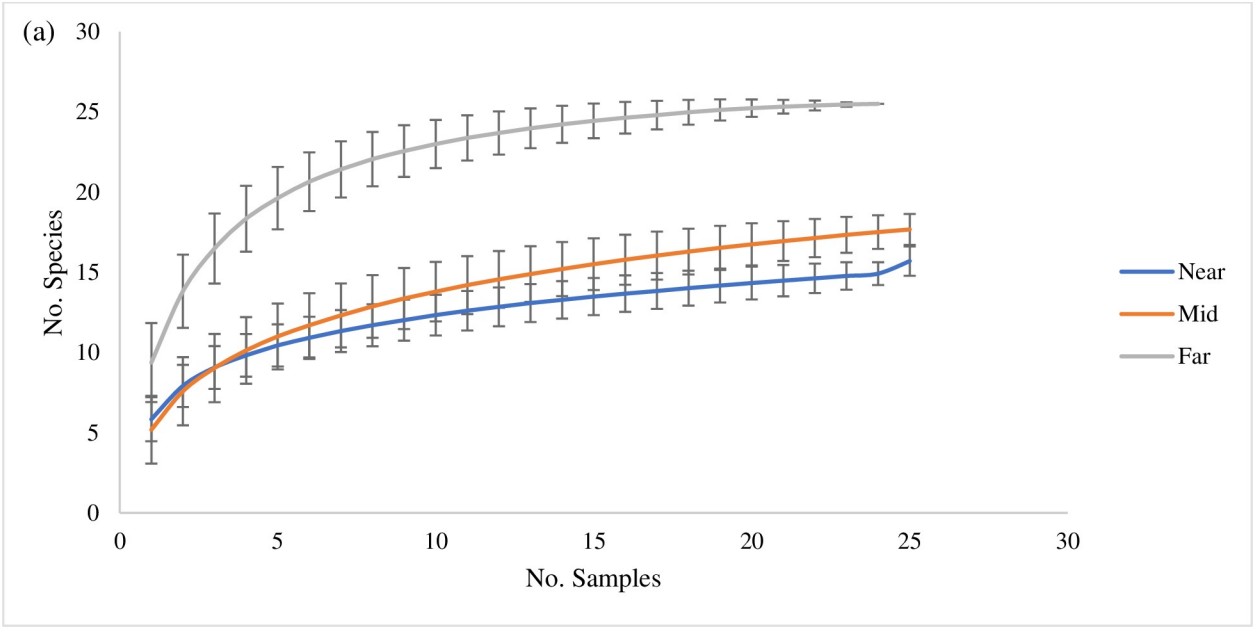

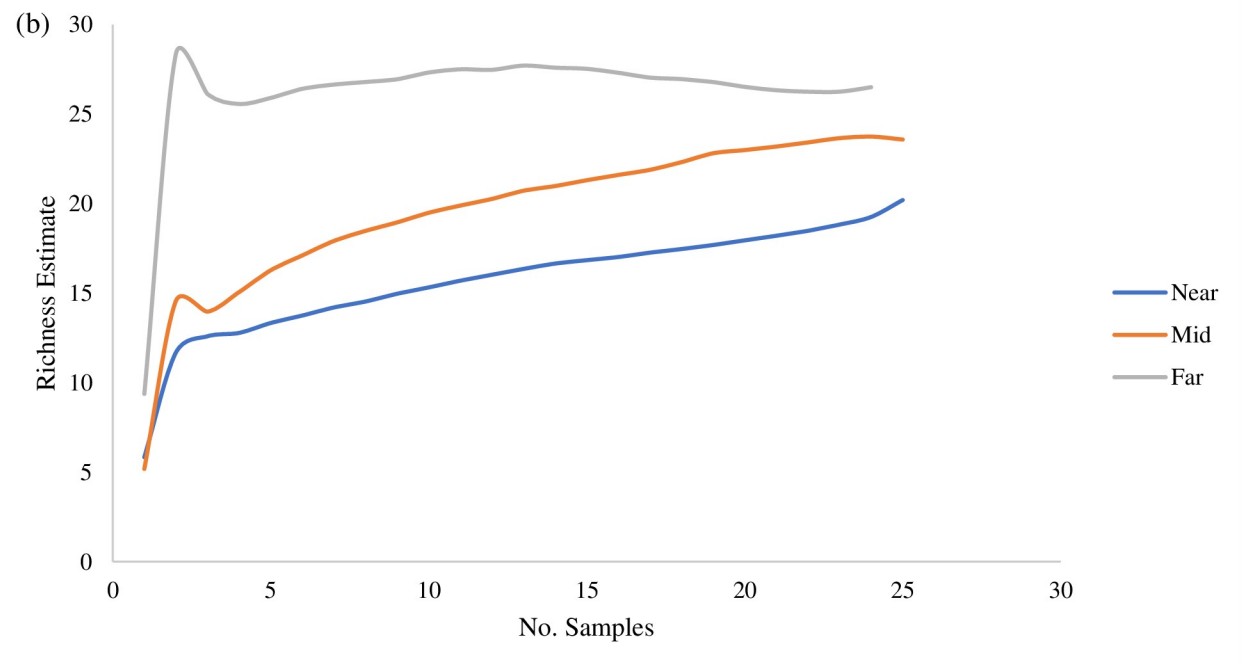

**Fig 2. Species accumulations curves and richness estimates.** (a) The species accumulation curve, *S* observed, shows the average observed species for each zone (near, mid and far) with standard deviation error bars present. (b) Average CHAO 1 richness estimator curves for each zone (near, mid and far). For species accumulation curves and CHAO 1 richness estimates of individual sites see Supporting Information.

particulate organic carbon (POC) flux (i.e. food supply) delivered, which can lead to higher densities of suspension-feeding organisms [52].Throughout most of the oceans, the most rapid rate of species turnover in the deep-sea occurs at the upper to mid-bathyal depths, the bathyal region is described as 200 to 4000 meters in this context [53]. Further research into the

**Table 2. Relative frequency of each taxon by site grouping.**

| Phylum | Lowest Known Taxon | Common Name | C1 & C2 (Far) | WD & GP (Mid, East Arm) | HK & TB (Mid, West Arm) | WTR & SILL (Near, East Arm) | J1 & J2 (Near, West Arm) |
|---|---|---|---|---|---|---|---|
| Porifera | Rosellidae | Glass sponge | 0.4167 | 0.5811 | 0.3793 | 0.5195 | 0.3500 |
| | *Aphrocallistes vastus* | Glass sponge | 0.0417 | 0.0405 | 0 | 0.0390 | 0.0125 |
| | Demospongiae | Demosponge | 0.1458 | 0.0135 | 0 | 0.0130 | 0.0125 |
| Cnidaria | *Primnoa pacifica (1)*[a] | Red Tree Coral | 0.3333 | 0.2297 | 0 | 0.0909 | 0.5625 |
| | *Primnoa pacifica (2)*[a] | Red Tree Coral | 0.3333 | 0.3784 | 0.0690 | 0.2078 | 0.6625 |
| | *Primnoa pacifica (3)*[a] | Red Tree Coral | 0.5208 | 0.4865 | 0.0948 | 0.2597 | 0.5125 |
| | *Primnoa pacifica (4)*[a] | Red Tree Coral | 0.2500 | 0.4595 | 0.8966 | 0.5844 | 0.2750 |
| | *Caryophyllia arnoldi* | Solitary cup coral | 0.5833 | 0.0270 | 0.1034 | 0.0000 | 0.5125 |
| | *Sarcodyction incrustans* | Encrusting stoloniferan coral | 0.7292 | 0.3378 | 0.1121 | 0.5974 | 0.7250 |
| | Hydrozoan turf | Hydrozoan turf | 0.2500 | 0.0270 | 0 | 0.1948 | 0.9875 |
| | *Cribrinopsis fernaldi* | Crimson anemone | 0.5625 | 0.6216 | 0.5172 | 0.6623 | 0.6625 |
| | *Metridium farcimen* | Giant white-plumed anemone | 0 | 0.0541 | 0.0172 | 0.0390 | 0 |
| | *Halipteris willemoesi* | Sea whip | 0.1250 | 0.0405 | 0 | 0 | 0 |
| Mollusca | *Fusitriton oregonensis* | Oregon triton snail | 0.3958 | 0.2568 | 0.0345 | 0 | 0 |
| | *Tritonia diomedea* | Pink tritonia Nudibranch | 0.5000 | 0.1622 | 0.0345 | 0.0130 | 0 |
| | *Akoya platinum* | Calliostomid snail | 0 | 0.0135 | 0 | 0 | 0 |
| | *Enteroctopus dofleini* | Giant Pacific Octopus | 0.0417 | 0.0135 | 0 | 0.0130 | 0 |
| | *Doryteuthis opalescens* | Opalescent inshore squid | 0 | 0 | 0 | 0 | 0.0125 |
| Annelida | Sabellidae | Feather duster worm | 0.0833 | 0.3514 | 0.0259 | 0.4416 | 0.2875 |
| Brachiopoda | *Laqueus californicus* | Lampshell Brachiopod | 0.8125 | 0.7568 | 0.6379 | 0.8831 | 0.9875 |
| Arthropoda | *Chirona evermanni* | Giant barnacle | 0.6667 | 0.7297 | 0.0086 | 0.0130 | 0 |
| | *Oregoniidae* sp. | Spider crab | 0 | 0 | 0.0345 | 0 | 0 |
| | *Chionoecetes* sp. | Snow crab | 0.0625 | 0.0541 | 0 | 0.0130 | 0 |
| | *Pandalus* spp. | Shrimp | 0 | 0.0405 | 0 | 0.0390 | 0.0500 |
| Echinodermata | *Hippasteria phrygiana* | Cushion star | 0.1042 | 0 | 0 | 0 | 0 |
| | *Gephyreaster swifti* | Gunpowder star | 0 | 0.0135 | 0 | 0 | 0 |
| | *Gorgonocephalus eucnemis* | Basket star | 0.3542 | 0.0541 | 0 | 0 | 0 |
| | *Ophiopholis aculeata* | Daisy brittle star | 0.5000 | 0 | 0 | 0 | 0 |
| | *Solaster dawsoni* | Morning sun star | 0.0625 | 0 | 0.0259 | 0 | 0 |
| | *Strongylocentrotus drobachiensis* | Green sea urchin | 0.1250 | 0.1216 | 0 | 0.0130 | 0.0750 |
| | *Psolus squamatus* | White creeping pedal sea cucumber | 0.3125 | 0.0811 | 0.0690 | 0 | 0 |
| | *Synallactes challengeri* | Challenger cucumber | 0.0417 | 0.0135 | 0 | 0 | 0 |
| | *Henricia* sp. | Blood star | 0.4583 | 0.1216 | 0.0345 | 0.0649 | 0 |

[a]*Primnoa pacifica* colonies were classified into size categories, each number described estimated colony heights as follows: (1) <0.25m, (2) 0.25–0.50m, (3) 0.50-1m, (4) >1m.

patterns of species turnover and drivers of changes in beta-diversity in GBNPP are strongly recommended.

It is expected that sampling effort (i.e. number of frames analyzed) would have an effect on the estimate of species richness but the analysis show that the species richness was highest at the study sites with the lowest number of frames analyzed (e.g. the Central Channel sites) and

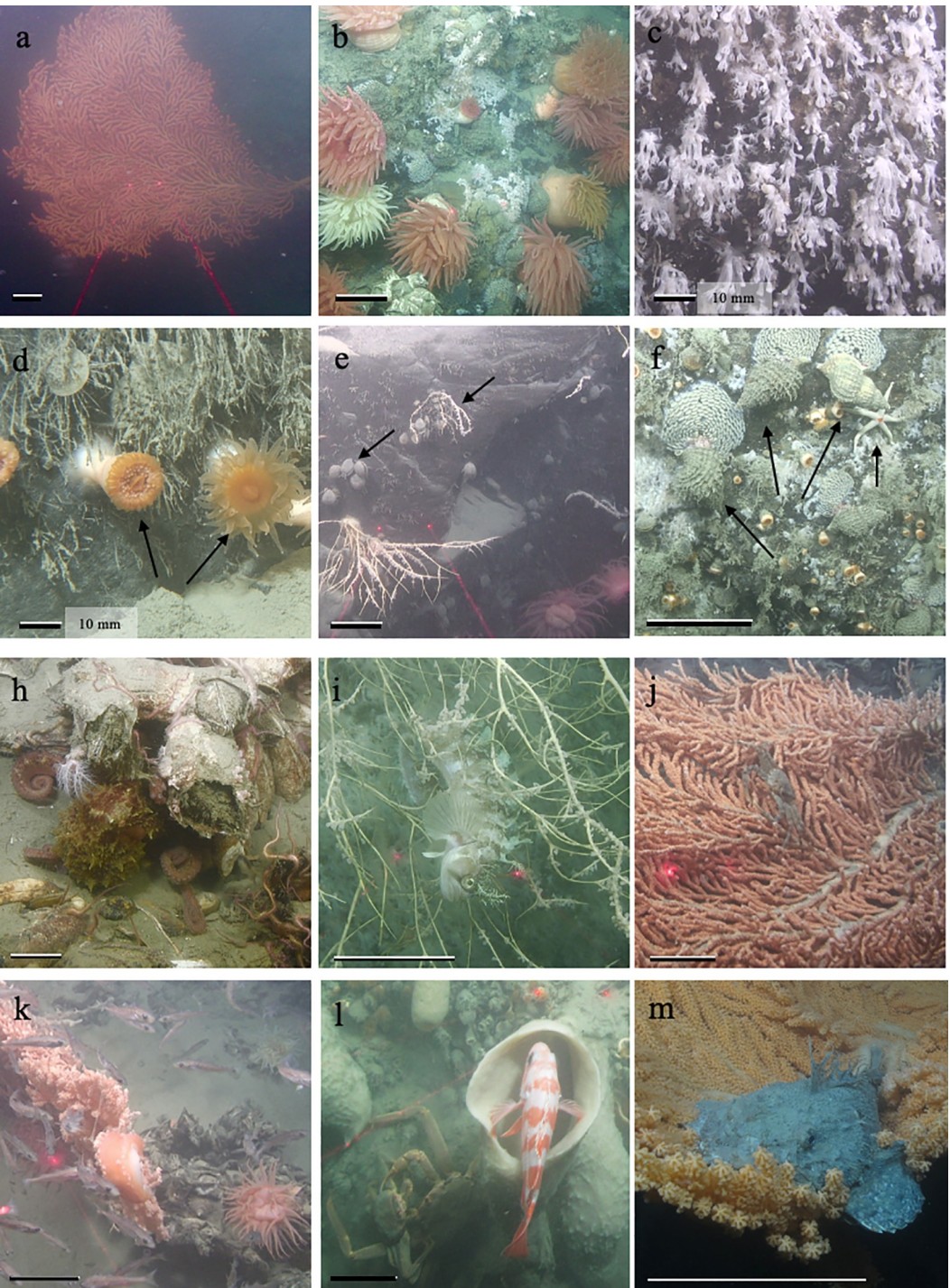

**Fig 3. Exemplar epifaunal taxa observed.** (a) large *Primnoa pacifica* colony (b) anemones *Cribrinopsis* sp. (c) encrusting stoloniferan coral *Sarcodyction incrustans* (d) close up of hydrozoan turf and arrows point to solitary cup coral *Caryophyllia arnoldi* (e) arrows point to small (<0.25 m) *P. pacifica* colony and brachiopods *Laqueus californicus* (f) arrows point to the snails *Fusitriton oregonensis* laying egg capsules as well as a blood star *Henricia* sp.–there are also *C. arnoldi* in this image. (h) Brittle star (*Ophiopholis* sp.) arms and an octopus, *Enteroctopus dofleini*, under barnacles, *Chirona evermanni* (i) decorated warbonnet *Chirolophis decorates* amongst branches of a *P. pacifica* colony (j) snow crab *Chionoecetes* sp. in a large *P. pacifica* colony (k) *P. pacifica* colony with an aggregation of juvenile Pacific cod *Gadus macrocephalus*, also in the image are barnacles (*C. evermanni*), an anemone (*Cribrinopsis* sp.), and the predatory nudibranch *Tritonia diodema* (l) red banded rockfish *Sebastes babcocki* over demospongiae and two snow crabs (*Chionoectes*sp.) (m) bigmouth sculpin *Hemitripterus bolini* stationary on edge of *P. pacifica* branches. Scale bars are 10cm unless otherwise noted.

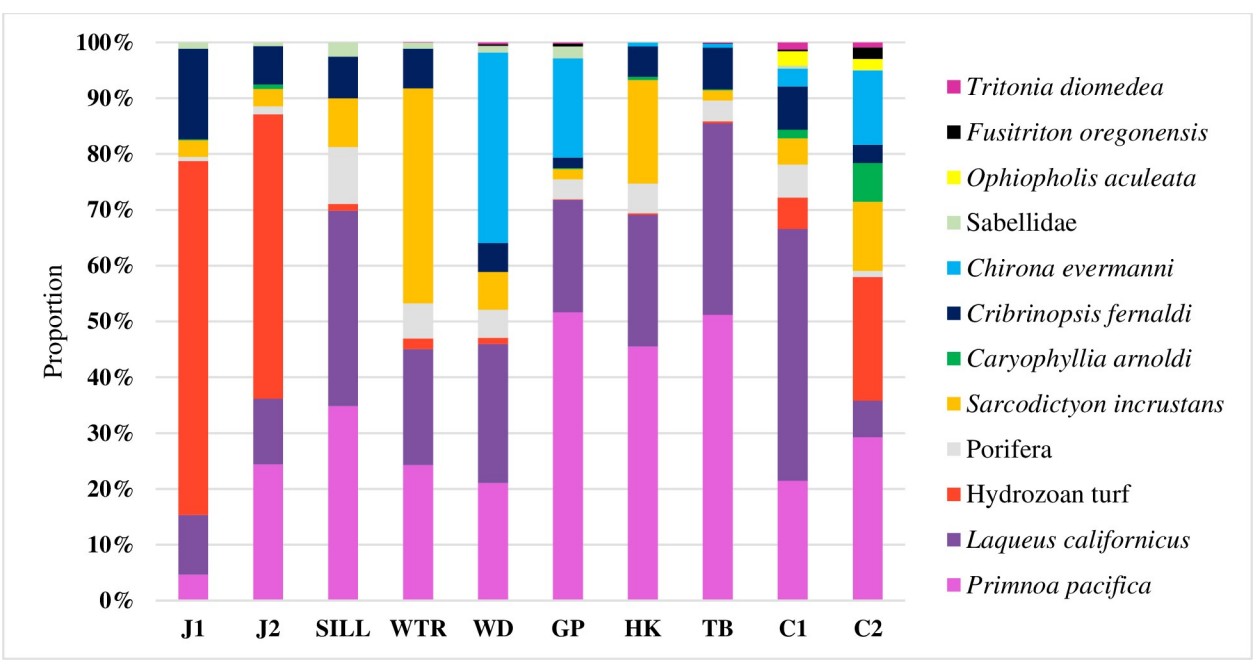

**Fig 4. Species dominance based on proportion at each site.** The sites in this graph are organized by proximity to glacial input (site J1 is closest to a tidewater glacier and site C2 is the furthest).

the sites with the highest number of frames analyzed had the lowest overall richness (Table 1). The high abundance of dense *P. pacifica* thickets at HK and TB in the West Arm led to more frames analyzed in an attempt to better observe the benthic community and to enhance our characterization of these sites. In a similar study of epibenthic assemblages on hard substrates in the North Sea, Michaelis et al. in 2019 found that taxon richness was highest in areas with the lowest image density and vice versa [54]. We suspect that increased sampling efforts across all study sites in GBNPP would support or even heighten the differences in community structure described herein. However, there is no foreseeable solution to the problem of large *P. pacifica* colonies obstructing the camera field-of-view of the substrate, particularly in areas of low water clarity, such as in glacial fjords.

Sites that were in geographical proximity to each other grouped together in the non-metric multidimensional scaling (nMDS) analysis. However, there was one notable exception, the species assemblages of the two Central Channel sites were different from one another (Fig 5). The higher-level geographical site groupings (i.e., near, mid, far) were shown to be significantly different from one another based on the results of the one-way ANOSIM routine, with a sample statistic (Global R) of 0.428 ($p = 0.018$). The sample statistic of 0.428 indicates that there were similarities of community assemblages but that there remained significant differences between site groupings. According to the similarity percentage routine (SIMPER), the taxa that contributed to approximately 50% of the dissimilarity between neighboring sites C1 and C2 were brachiopods, hydrozoan turf and barnacles. Site C1 was dominated by brachiopods and sponges, and site C2 was dominated by hydrozoan turf and barnacles. The sites were separated by 1.5 kilometers latitudinally and the transects at both sites were conducted between 200 and 300 meters. Due to the proximity of the two sites, it is reasonable to assume that oceanographic variables affecting them are comparable, therefore it should be noted that the difference in underlying geology could be a driver of the observed megafaunal differences.

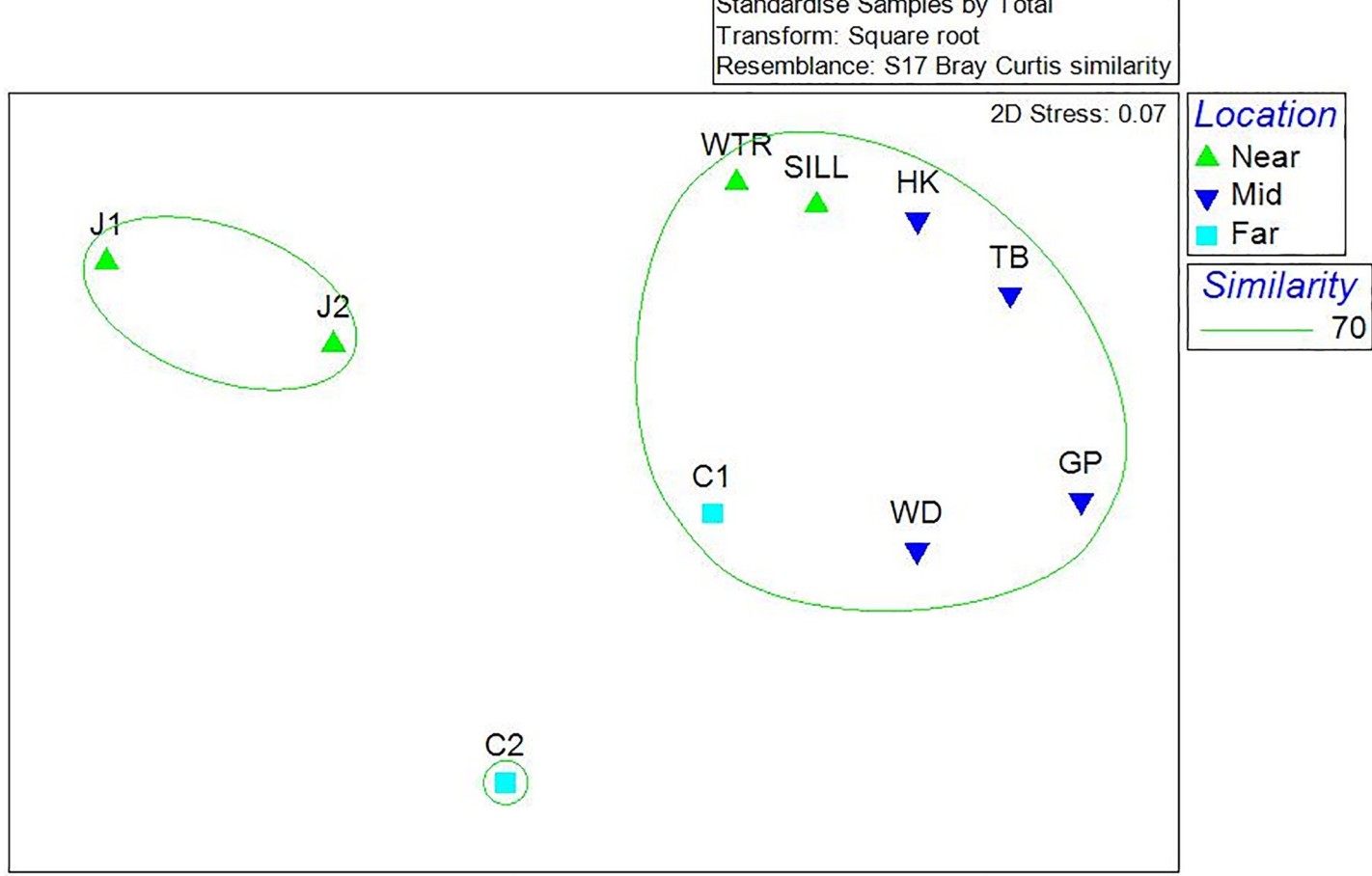

**Fig 5. Non-metric multidimensional scaling plot.** This graph represents the nMDS analysis in two-dimensional space. The nMDS used a Bray-Curtis similarity matrix calculated from percent coverage data that were standardized and square-root transformed. Green circles indicate 70% similarity.

The complete-linkage hierarchical cluster dendrogram showed three groupings of similar species composition: the Johns Hopkins sites (J1 and J2), the Central Channel site that was furthest from glaciers (C2) and the rest of the sites (C1, WTR and SILL, HK and TB, WD and GP) (Fig 6). The secondary cluster showed similarities between East Arm sites (WD and GP), the West Arm sites (HK and TB), and the near glacier sites in the East Arm (WTR and SILL) (Fig 6). This suggests a possible pattern of fidelity in species composition determined not only by their proximity to glaciers but also by which fjord they inhabit, although further studies are necessary to reinforce the presence of such pattern. Two-way nested ANOSIM did not show significance of dissimilarity between sites based on which fjord the sites were in, but because of the small number of sites, not enough replicates (therefore permutations) were possible to allow a reasonable significance test [55]. The differences in benthic community composition between fjords deserves further study as the two arms of Glacier Bay have different rates of glacial retreat, freshwater input, basin morphology and flow dynamics [14, 31].

Etherington et al. in 2007 found that the highest levels of chlorophyll *a* in Glacier Bay were in the central bay and the lower reaches of the East and West Arms [31]. The water column conditions–low stratification, low sedimentation, and moderate current speeds–in these locations were also the most optimal for a high concentration of benthic organisms. Sedimentation is a strong control on species diversity and distribution, and sedimentation levels near

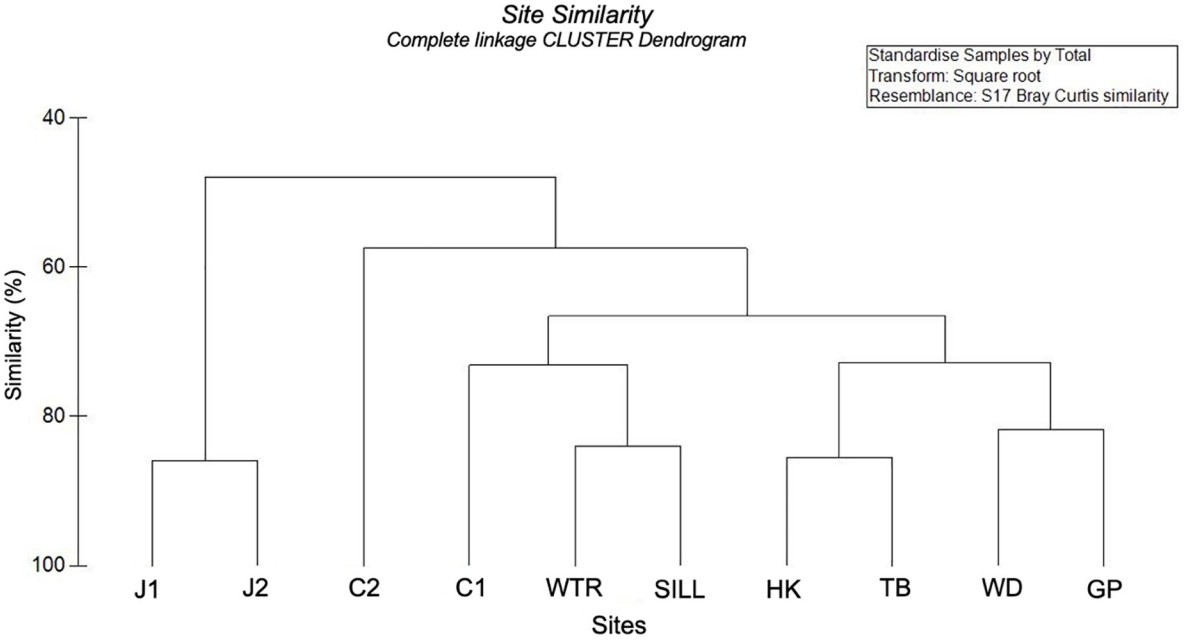

**Fig 6. Site similarity dendrogram.** This complete-linkage CLUSTER analysis dendrogram was constructed from a Bray-Curtis similarity matrix calculated from percent coverage data, which were standardized and square-root transformed for analysis.

tidewater glaciers can be some of the highest in the world [44, 56]. These oceanographic patterns support our findings of lower diversity and abundance at near-glacier sites, and those of higher diversity and abundance in the lower East and West Arms as well as in the Central Channel.

Sites that were in the same geographical zone (near, mid, far) had strong similarities of community composition. The SIMPER routine demonstrated that the average similarity within the near zone sites was 70.63%, average similarity within mid-fjord zone sites was 78.64%, and average similarity within far zone sites was 72.31%. *Primnoa pacifica* at the J1 site were all size class 1 and 2 colonies, whereas at HK and TB, colonies recorded over 1m in size (class 4) represented more than 90% and 70% respectively. The lack of larger sizes classes being observed at J1 (and few at J2) is likely owing to the proximity of the glacier, where substrates have only recently been exposed for colonization (J1 is only 4km from the glacier), and in addition enhanced stressors from glacial inputs potentially reducing growth rates. The lack of small colonies at HK and TB could be attributed to low visibility of substrates surrounding larger colonies skewing size class data. Sites in the East Arm and Central Channel (C1 and C2) had a more mixed representation of size classes, with relatively even proportions being seen at all populations except the sill, where over 70% of colonies were in the largest size class (S4 Fig). The second largest driver of similarity was the brachiopod *Laqueus californicus* (S1 Table). The near-glacier and far sites had an average similarity of 65.09%, which was largely driven by the abundance of hydrozoan turfs at the near-glacier sites, and the abundance of barnacles and brachiopods at the far sites. In addition, the triton snail, *Fusitriton oregonensis*, and ophiuroids were observed at the far sites and not at the near sites. The near sites in the West Arm, J1 and J2 were dominated by hydrozoan turfs while the near sites in the East Arm, WTR and SILL, were dominated by brachiopods and the encrusting stoloniferan coral *Sarcodictyon incrustans*.

Substrate type likely contributed to the differences in species composition observed at the two Central Channel sites, as well as processes not able to be quantified by this study (e.g. local

or meso-scale flow dynamics, food availability, predation, competition, etc.). The southern Central Channel site (C2), was characterized by the presence of large barnacle outcrops and dense assemblages of anemones and ophiuroids. Many ophiuroids, basket stars and brittle stars, live in mutualistic relationships with large structure-forming corals [8, 57]. They use their perch on coral branches to more easily access food in the water column and sometimes even remove suspended materials that could suffocate coral polyps [57]. Their presence in the Central Channel and lower East and West Arms of Glacier Bay are likely due to the increased tidal currents in those areas of the fjord system [31]. The northern Central Channel site (C1) transect covered expanses of both soft sediment (silt) bottom where pennatulaceans were observed, and hard substrate that had dense populations of the solitary scleractinian coral *Caryophyllia arnoldi* and brachiopods. Brachiopods occur frequently in Chilean [58] and British Columbian [59, 60] fjords, as well as in the fjords of Southeast Alaska (Stone and Mondragon 2018). Tunnicliffe and Wilson in 1988 documented that the endemic brachiopod species, *L. californicus*, is tolerant to high turbidity, high turbulence, and low oxygen concentration environments [60]. Thayer in 1985 demonstrated that brachiopods are not a palatable prey item [61]. The lack of predation on brachiopods and their ability to succeed in marginal environments lends to their ubiquity in Pacific fjords.

Shelter seeking fish (including ambush predators in the family Scorpaeniformes), crabs and shrimp (Fig 3I, 3J and 3M) were observed at the far and mid fjord sites. The majority (>75%) of mobile taxa were observed in frames with *P. pacifica* and sponges. At these sites, the presence of the predatory nudibranch *Tritonia diomedea* (Fig 3K) was recorded on *P. pacifica* branches as was the presence of the triton snail *F. oregonensis*. The triton snails were observed laying egg capsules on bedrock adjacent to *P. pacifica* colonies (Fig 3F), and three distinct types of unidentified egg masses were observed on coral branches. Stone and Mondragon in 2018 suggested that *P. pacifica* demonstrates pioneering species characteristics due to its presence on substrate that has been deglaciated for as little as two decades [14]. The relative homogeneity and paucity of species at higher latitude sites indicate that they might also be pioneering species [36]. Another notable pattern is the scarcity of higher trophic level predators and shelter-seeking taxa at the near-glacier sites, this is potentially due to the lack of large coral colonies in these areas, resulting in reduced prey availability.

*P. pacifica* is classified as Essential Fish Habitat (EFH) in the eastern North Pacific [38] and is protected by provisions in the Sustainable Fisheries Act [62]amendment to the Magnuson-Stevens Act [63] as Habitat Areas of Particular Concern (HAPC) in the Gulf of Alaska [41, 64]. Deep sea coral and sponge habitats in Alaska function as important biogenic structures that support diverse communities of invertebrates, which in turn may support economically important species of fish and crabs [11]. The influence of *P. pacifica* on its surrounding community is significant [41]. Although this research targeted *P. pacifica* communities in GBNPP and did not sample unstructured substrate, we suggest that our results support the contention that *P. pacifica* is important biogenic habitat in deep-sea ecosystems.

*Primnoa pacifica* has a high potential for physical disturbance due to its arboreal morphology. In GBNPP specifically, disturbances include iceberg scour, and rock and ice slides due to the steep fjord walls. These natural physical disturbances are in addition to the anthropogenic disturbances caused by climate change. The health of coral populations is critical to the diversity and biomass of associated species [8, 11, 13, 41]. Determining the potential for resilience could have important implications for the conservation of these cold-water coral habitats, especially for those that are not protected from anthropogenic disturbances. GBNPP is a model environment in which to investigate such questions because of the protections afforded to it since 1925. This study is a natural experiment that results in the identification of patterns based on differences in physiographic settings and patterns of natural disturbance. These

results identify variations in community structure that could be expected in other areas and inform the expectations of recovery from natural or human caused disturbances.

Research on ecological recovery and resilience is already the focus of much of the terrestrial and aquatic research that takes place in GBNPP due to the unique and varying rates of glacial recession. The patterns of diversity and abundance described here demonstrate a gradient of species composition that largely correspond to latitude and glacial influence, demonstrating the general patterns of a fjord diversity cline. Studying the processes–such as flow dynamics, fjord hydrology, larval dispersal, recruitment, predation, competition, etc.–that drive the patterns described herein, is critical to the conservation of these ecosystems.

## Conclusions

This is the first study to report on cold-water coral community structural analysis within National Park boundaries, as well as the first description of bathyal benthic community structure in Glacier Bay National Park and Preserve, Alaska. This study found that cold-water coral communities were generally more diverse and abundant as the distance from glacial input increased. Glacial fjords are effectively living laboratories for deep-sea biologists, providing the unique opportunity to study the deep-sea in an accessible and relatively controlled environment.

## Supporting information

**S1 Fig. Species accumulation curves for sites in near zones.** (a) The species accumulation curve, *S* observed, shows the average observed species for each site in the Near zone (J1, J2, Sill, WTR) with standard deviation error bars present. (b) Average CHAO 1 richness estimator curves for each site in the Near zone (J1, J2, Sill, WTR) with standard deviation error bars present.
(DOCX)

**S2 Fig. Species accumulation curves for sites in mid zones.** (a) The species accumulation curve, *S* observed, shows the average observed species for each site in the Mid zone (WD, HK, GP, TB) with standard deviation error bars present. (b) Average CHAO 1 richness estimator curves for each site in the Mid zone (WD, HK, GP, TB) with standard deviation error bars present.
(DOCX)

**S3 Fig. Species accumulation curves for sites in far zones.** (a) The species accumulation curve, *S* observed, shows the average observed species for each site in the Far zone (C1, C2) with standard deviation error bars present. (b) Average CHAO 1 richness estimator curves for each site in the Far zone (C1, C2) with standard deviation error bars present.
(DOCX)

**S4 Fig. Frequency of *Primnoa pacifica* size class occurrence at each site.**
(DOCX)

**S1 Table. Taxon contribution to similarity between sites–SIMPER analysis.**
(DOCX)

## Acknowledgments

We would like to acknowledge the captain and crew of the MV Norseman II, the pilots and technicians of the Kraken II ROV, and the diverse members of the science party who all

contributed to the success of the expedition. Kevin Joy (University of Connecticut) deserves a particular note of thanks for putting all of the ship and vehicle elements of the cruise together. I would like to acknowledge Damian Brady, Robert Sone and Brenda Hall for their help with data analysis, species identification, and all aspects of glaciology. Lisa Etherington, Craig Murdoch, Lewis Sharman, and the GBNPP staff, whose contributions were essential to the success of this research expedition. Lastly to Bill Favitta, for making a map of GBNPP in GIS for this manuscript.

## Author Contributions

**Conceptualization:** Rhian G. Waller, Peter J. Auster.

**Data curation:** Élise C. Hartill, Rhian G. Waller.

**Formal analysis:** Élise C. Hartill.

**Funding acquisition:** Rhian G. Waller, Peter J. Auster.

**Investigation:** Élise C. Hartill.

**Methodology:** Élise C. Hartill, Rhian G. Waller, Peter J. Auster.

**Project administration:** Rhian G. Waller.

**Resources:** Rhian G. Waller, Peter J. Auster.

**Software:** Rhian G. Waller, Peter J. Auster.

**Supervision:** Rhian G. Waller, Peter J. Auster.

**Validation:** Rhian G. Waller, Peter J. Auster.

**Visualization:** Élise C. Hartill.

**Writing – original draft:** Élise C. Hartill.

**Writing – review & editing:** Élise C. Hartill, Rhian G. Waller, Peter J. Auster.

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
