## [Decision Letter · Decision Letter 0]

8 Apr 2020

PONE-D-20-05571

Deep benthic coral habitats of Glacier Bay National Park and Preserve, Alaska

PLOS ONE

Dear Ms. Hartill,

Thank you for submitting your manuscript to PLOS ONE. After careful consideration, we feel that it has merit but does not fully meet PLOS ONE’s publication criteria as it currently stands. Therefore, we invite you to submit a revised version of the manuscript that addresses the points raised during the review process.

Thanks for submitting your manuscript to PLoS One. I feel that the findings of the two reviewers warrants a decision of major revision. I note particularly that there is significant overlap with a priorly published thesis, which naturally happens, but is a gray area for some peer reviewed literature. However, I do hope that you take on board the insightful reviewers comments about improving the manuscript as suggested. In doing so, should lead to a far stronger submission that is more distinct from the thesis.

We would appreciate receiving your revised manuscript by May 23 2020 11:59PM. To enhance the reproducibility of your results, we recommend that if applicable you deposit your laboratory protocols in protocols.io, where a protocol can be assigned its own identifier (DOI) such that it can be cited independently in the future. For instructions see: http://journals.plos.org/plosone/s/submission-guidelines#loc-laboratory-protocols

We look forward to receiving your revised manuscript.

Kind regards,

Andrew Davies

Academic Editor

PLOS ONE

Journal Requirements:

2) PLOS requires an ORCID iD for the corresponding author in Editorial Manager on papers submitted after December 6th, 2016. Please ensure that you have an ORCID iD and that it is validated in Editorial Manager. To do this, go to ‘Update my Information’ (in the upper left-hand corner of the main menu), and click on the Fetch/Validate link next to the ORCID field. This will take you to the ORCID site and allow you to create a new iD or authenticate a pre-existing iD in Editorial Manager. Please see the following video for instructions on linking an ORCID iD to your Editorial Manager account: https://www.youtube.com/watch?v=_xcclfuvtxQ

3) Please include captions for your Supporting Information files at the end of your manuscript, and update any in-text citations to match accordingly. Please see our Supporting Information guidelines for more information: http://journals.plos.org/plosone/s/supporting-information.

4) We note that [Figure 1] in your submission contain [map/satellite] images which may be copyrighted. All PLOS content is published under the Creative Commons Attribution License (CC BY 4.0), which means that the manuscript, images, and Supporting Information files will be freely available online, and any third party is permitted to access, download, copy, distribute, and use these materials in any way, even commercially, with proper attribution. For these reasons, we cannot publish previously copyrighted maps or satellite images created using proprietary data, such as Google software (Google Maps, Street View, and Earth). For more information, see our copyright guidelines: http://journals.plos.org/plosone/s/licenses-and-copyright.

i.    You may seek permission from the original copyright holder of Figure(s) [#] to publish the content specifically under the CC BY 4.0 license.

ii.    If you are unable to obtain permission from the original copyright holder to publish these figures under the CC BY 4.0 license or if the copyright holder’s requirements are incompatible with the CC BY 4.0 license, please either i) remove the figure or ii) supply a replacement figure that complies with the CC BY 4.0 license. Please check copyright information on all replacement figures and update the figure caption with source information. If applicable, please specify in the figure caption text when a figure is similar but not identical to the original image and is therefore for illustrative purposes only.

Reviewers' comments:

Reviewer's Responses to Questions

**Comments to the Author**

1. Is the manuscript technically sound, and do the data support the conclusions?

Reviewer #1: Partly

Reviewer #2: Partly

2. Has the statistical analysis been performed appropriately and rigorously? 

Reviewer #1: Yes

Reviewer #2: Yes

3. Have the authors made all data underlying the findings in their manuscript fully available?

Reviewer #1: Yes

Reviewer #2: Yes

4. Is the manuscript presented in an intelligible fashion and written in standard English?

Reviewer #1: Yes

Reviewer #2: Yes

5. Review Comments to the Author

Reviewer #1: It was a pleasure to review this article as it highlights an area of research (e.g. fjord habitats) that still has a limited knowledge base. The manuscript provides a good overview of Primnoa pacifica communities within 10 study sites in the fjords within Glacier Bay National Park and Reserve in Alaska. It investigates community changes within these coral communities in relation to the proximity of glacial heads and shows that richness is greatest further away from the glacial heads.

In its current state, the manuscript reads like a thesis and needs improvement. The article is generally well-written, though there are areas that are disorganized and difficult to follow, particularly in the methods and results/discussion. The article lacks a true objective. The statistical analysis needs more justification and explanation, though I do not believe re-analysis is necessarily needed. The data supports that the communities furthest away from the glaciers has a higher richness, however the author’s suggestion that Primnoa pacifica is important to its surrounding community cannot necessarily be supported by the data since the authors did not examine areas where P. pacifica was absent. In addition to what has already been stated, I have selected some major points that should be addressed, and highlighted more specific points below.

The main issues of the manuscript are as follows:

• The objective of the study was not clearly stated, though if I am not mistaken, the main objective was to examine changes in fjord-based coral communities in proximity to the glacial input. However, the lack of clear objectives made understanding the purpose of the methods rather difficult.

• I think subheadings would greatly help focus the MS more and possibly help the authors match the subheadings to their objectives.

• I found that there was information presented in the results that was not clearly explained in the methods. The statistical analysis section lists a variety of statistical techniques without providing a real justification as to why they were done, other than a short explanation.

• I think there is a lack of literature about fjord communities and their physical properties in the introduction. While I understand that literature is limited in this topic, there are some good articles about glaciated fjord systems in Svalbard and cold-water coral communities within Norwegian fjords. I think providing more specific examples from the literature rather than just stating that deep-water corals or Primnoa pacifica has a significant influence on its community would be very beneficial for the MS.

• I think the figures could use some work. If colors are going to be used, make sure they are consistent between figures because as of right now almost every figure has a different color scheme when referring to near, mid, and far stations. Text should also be consistent between figures. Stick to one font style (and color), and decide whether or not station numbers or site names will be used. Please ensure that font size is large enough (particularly fig. 1).

Some additional points that should be addressed:

• Please check species names. Make sure the name and authority are correct on WoRMs (or any other up to date taxonomic database). If the taxa were only able to be identified to genus level, make sure to include sp. or spp. (without italics on the sp./spp.). Use brackets on authorities only if WoRMs (or other taxonomic database) uses it; brackets indicate that the species have been moved from the original genus. Also, I think table 2 should have a more robust classification system, for example, I have no idea what is meant by “cucumber, sediment”.

• A more detailed site description would be extremely beneficial to the MS. This includes sill heights, basin depths, and water mass structure within the fjords.

Specific points:

Abstract:

1. Abstract does not provide a full overview of the methods other than that a ROV surveyed the fjord. It would be good to include more information regarding the survey design (e.g. number of sites surveyed, whether environmental data was collected and for what purpose, etc).

2. The objective of the study is not clearly mentioned in the abstract.

Introduction:

3. Lines 73-84: it would be good to know the physical properties of the fjord and fjord arms since these are the study sites for the MS. This includes basin depth, sill depth, total length, water mass structure.

4. Line 119: When starting a sentence with a species name, it should be written in full.

5. Lines 126-134: It is difficult to identify the objective(s) of the study here. Is it about identifying how glacier distance impacts P. pacifica colony and surrounding megabenthic communities, or how the presence of P. pacifica influences the associated megabenthic community? Or is it something else? I suggest clearly writing out the objectives of the study are here.

6. Lines 128: Why were 10 study sites selected for this study? Were these study sites selected based on the 16 deep-water sites surveyed in 2010 (mentioned earlier in the MS)?

Materials and Methods:

7. While the methods were concise, I found that they were a bit disorganized and difficult to follow. Subheadings would greatly help this section and keep specific or related information together.

8. Line 141: I am not sure what "inference based on habitat knowledge from previous studies" means. Please clarify. Was this from the 2010 survey?

9. Line 153: CTD should be fully written out before using the abbreviation.

10. Line 162: How was the substrate type determined?

11. Lines 173-179: Was there a size cut-off for the species that were enumerated? Many video surveys exclude fauna that are smaller than 1 cm, was that the case here?

12. Lines 180: Size classes of P. pacifica should be included here.

13. Lines 181: How were new individuals determined? What was the purpose of collecting information about the new individuals (and size classes)?

Results and Discussion:

14. Similar to the methods, I think that subheadings should be included here as well. This results/discussions presented did not match the order their corresponding methods were presented.

15. I noticed that there was some inclusion of results (and corresponding methods) that were not clearly presented or justified in the methods section (such as the sudden mention of a two-way nested ANOSIM on the sites based on which fjord the sites were in). Make sure every method that corresponds with the presented results had been clearly mentioned and justified in the methods section.

16. I think the total area surveyed or the total number of quadrats should be included in the results.

17. Line 204: How was turbidity determined in the survey? There is nothing about it mentioned in the methods.

18. Lines 385-388: I think it is difficult to say that all benthic taxa were centered around colonies of P. pacifica since this study only really focused on areas with P. pacifica presence. I think this statement would be more founded if the study also included areas with P. pacifica absence.

Tables and Figures:

Table 1:

19. Table gives the approximate area surveyed. Does this include the total length of the transects or just the total area of the analysed quadrats?

Table 2:

20. I think a more robust classification system should be used here. For example, what is "cucumber, sediment" referring to?

21. Check species names. Make sure genus only names has sp. or spp. (not italicized).

22. Why are fish not included here since other mobile taxa are included?

Figure 1:

23. This figure needs a lot of improvement. The font on this figure is rather small and should be improved. The scale bar is barely visible. The icons in the legend should be lined up nicely. It would also be helpful to have the site names included.

Figure 3 and 7:

24. Why are these two figures separated since figure 3 also has mobile taxa? They are both very nice but could be combined.

Supplementary material:

Figure 1 and 2:

25. Please check the site names for this figure. In 1a and 2a the station names are included, but in 1b and 2b the station numbers are included instead. Be consistent and just use site names or station number throughout the entire text. Additionally, the colors corresponding the station numbers in Figure 1 also seem to be flipped for Sill (station 13 according to table 1) and WTR (station 4 according to table 1).

Reviewer #2: This work is worthy of publication – it focuses on important benthic habitats that provide Essential Fish Habitat and support keystone species. The study is focused on a high latitude ecosystem that is susceptible to climate change, and it is valuable to understand current species’ distributions from which to measure future change. However, I have a number of comments and concerns that I think need to be addressed before this is suitable for publication. I therefore recommend publication with major revisions.

6. PLOS authors have the option to publish the peer review history of their article (what does this mean?). If published, this will include your full peer review and any attached files.

Reviewer #1: Yes: Heidi Kristina Meyer

Reviewer #2: No

---

## [Author Response · Author response to Decision Letter 0]

13 Jul 2020

Comments to the Author

1. Is the manuscript technically sound, and do the data support the conclusions?

Reviewer #1: Partly

Reviewer #2: Partly

2. Has the statistical analysis been performed appropriately and rigorously? 

Reviewer #1: Yes

Reviewer #2: Yes

3. Have the authors made all data underlying the findings in their manuscript fully available?

Reviewer #1: Yes

Reviewer #2: Yes

4. Is the manuscript presented in an intelligible fashion and written in standard English?

Reviewer #1: Yes

Reviewer #2: Yes

5. Review Comments to the Author

Reviewer #1: It was a pleasure to review this article as it highlights an area of research (e.g. fjord habitats) that still has a limited knowledge base. The manuscript provides a good overview of Primnoa pacifica communities within 10 study sites in the fjords within Glacier Bay National Park and Reserve in Alaska. It investigates community changes within these coral communities in relation to the proximity of glacial heads and shows that richness is greatest further away from the glacial heads.

In its current state, the manuscript reads like a thesis and needs improvement. The article is generally well-written, though there are areas that are disorganized and difficult to follow, particularly in the methods and results/discussion. The article lacks a true objective. The statistical analysis needs more justification and explanation, though I do not believe re-analysis is necessarily needed. The data supports that the communities furthest away from the glaciers has a higher richness, however the author’s suggestion that Primnoa pacifica is important to its surrounding community cannot necessarily be supported by the data since the authors did not examine areas where P. pacifica was absent. In addition to what has already been stated, I have selected some major points that should be addressed, and highlighted more specific points below.

The main issues of the manuscript are as follows:

• The objective of the study was not clearly stated, though if I am not mistaken, the main objective was to examine changes in fjord-based coral communities in proximity to the glacial input. However, the lack of clear objectives made understanding the purpose of the methods rather difficult.

- Corrected in manuscript, line 273 “1.4 Objective” in “Revised Article with Changes Highlighted” file

• I think subheadings would greatly help focus the MS more and possibly help the authors match the subheadings to their objectives.

- Corrected in manuscript

• I found that there was information presented in the results that was not clearly explained in the methods. The statistical analysis section lists a variety of statistical techniques without providing a real justification as to why they were done, other than a short explanation.

- Corrected in manuscript, addressed in section “2.4 Statistical Analysis” 

• I think there is a lack of literature about fjord communities and their physical properties in the introduction. While I understand that literature is limited in this topic, there are some good articles about glaciated fjord systems in Svalbard and cold-water coral communities within Norwegian fjords. I think providing more specific examples from the literature rather than just stating that deep-water corals or Primnoa pacifica has a significant influence on its community would be very beneficial for the MS.

- Corrected in manuscript, lines 89-112 in “Revised Article with Changes Highlighted” file

• I think the figures could use some work. If colors are going to be used, make sure they are consistent between figures because as of right now almost every figure has a different color scheme when referring to near, mid, and far stations. Text should also be consistent between figures. Stick to one font style (and color), and decide whether or not station numbers or site names will be used. Please ensure that font size is large enough (particularly fig. 1).

- Corrected in manuscript

Some additional points that should be addressed:

• Please check species names. Make sure the name and authority are correct on WoRMs (or any other up to date taxonomic database). If the taxa were only able to be identified to genus level, make sure to include sp. or spp. (without italics on the sp./spp.). Use brackets on authorities only if WoRMs (or other taxonomic database) uses it; brackets indicate that the species have been moved from the original genus. Also, I think table 2 should have a more robust classification system, for example, I have no idea what is meant by “cucumber, sediment”.

- Corrected in manuscript

• A more detailed site description would be extremely beneficial to the MS. This includes sill heights, basin depths, and water mass structure within the fjords.

- Corrected in manuscript, lines 154-228 in “Revised Article with Changes Highlighted” file

Specific points:

Abstract:

1. Abstract does not provide a full overview of the methods other than that a ROV surveyed the fjord. It would be good to include more information regarding the survey design (e.g. number of sites surveyed, whether environmental data was collected and for what purpose, etc).

- Corrected in manuscript

2. The objective of the study is not clearly mentioned in the abstract.

- Corrected in manuscript

Introduction:

3. Lines 73-84: it would be good to know the physical properties of the fjord and fjord arms since these are the study sites for the MS. This includes basin depth, sill depth, total length, water mass structure.

- Corrected in manuscript, lines 154-228 in “Revised Article with Changes Highlighted” file

4. Line 119: When starting a sentence with a species name, it should be written in full.

- Corrected in manuscript

5. Lines 126-134: It is difficult to identify the objective(s) of the study here. Is it about identifying how glacier distance impacts P. pacifica colony and surrounding megabenthic communities, or how the presence of P. pacifica influences the associated megabenthic community? Or is it something else? I suggest clearly writing out the objectives of the study are here.

- Corrected in manuscript, section “1.4 Objective” 

6. Lines 128: Why were 10 study sites selected for this study? Were these study sites selected based on the 16 deep-water sites surveyed in 2010 (mentioned earlier in the MS)?

- Corrected in manuscript in “2.1 Site Selection”

Materials and Methods:

7. While the methods were concise, I found that they were a bit disorganized and difficult to follow. Subheadings would greatly help this section and keep specific or related information together.

- Corrected in manuscript

8. Line 141: I am not sure what "inference based on habitat knowledge from previous studies" means. Please clarify. Was this from the 2010 survey?

- Corrected in manuscript, clarified line 285-286 in “Revised Article with Changes Highlighted” file

- 

9. Line 153: CTD should be fully written out before using the abbreviation. 

- Corrected in manuscript

10. Line 162: How was the substrate type determined?

- Corrected in manuscript, Wentworth scale (Holme and McIntyre 1971) line 505 in “Revised Article with Changes Highlighted” file

11. Lines 173-179: Was there a size cut-off for the species that were enumerated? Many video surveys exclude fauna that are smaller than 1 cm, was that the case here?

- Corrected in manuscript, line 650

12. Lines 180: Size classes of P. pacifica should be included here.

- Corrected in manuscript, lines 661-665

13. Lines 181: How were new individuals determined? What was the purpose of collecting information about the new individuals (and size classes)?

- Corrected in manuscript, lines 666-688

Results and Discussion:

14. Similar to the methods, I think that subheadings should be included here as well. This results/discussions presented did not match the order their corresponding methods were presented. 

- Corrected in manuscript

15. I noticed that there was some inclusion of results (and corresponding methods) that were not clearly presented or justified in the methods section (such as the sudden mention of a two-way nested ANOSIM on the sites based on which fjord the sites were in). Make sure every method that corresponds with the presented results had been clearly mentioned and justified in the methods section.

- Corrected in manuscript

16. I think the total area surveyed or the total number of quadrats should be included in the results.

- Corrected in manuscript

17. Line 204: How was turbidity determined in the survey? There is nothing about it mentioned in the methods.

- Turbidity was only visually assessed qualitatively, addressed in manuscript

18. Lines 385-388: I think it is difficult to say that all benthic taxa were centered around colonies of P. pacifica since this study only really focused on areas with P. pacifica presence. I think this statement would be more founded if the study also included areas with P. pacifica absence.

- Corrected in manuscript, clarified in “2.2 Survey Method”

Tables and Figures:

Table 1:

19. Table gives the approximate area surveyed. Does this include the total length of the transects or just the total area of the analysed quadrats?

- Corrected in manuscript, replaced “transect start depth” with “transect length” instead & “area surveyed” accounts for the sum of quadrats surveyed

Table 2:

20. I think a more robust classification system should be used here. For example, what is "cucumber, sediment" referring to?

- Corrected in manuscript, I think I mistakenly added an older table that didn’t have species names to this manuscript

21. Check species names. Make sure genus only names has sp. or spp. (not italicized).

- Corrected in manuscript

22. Why are fish not included here since other mobile taxa are included?

- Encounter rates for fish were low overall and many species were transient (i.e., not necessarily associated with coral habitats per se). Because inclusion of fish would have required some a priori assumptions that appeared to us to be much too deterministic of outcome, we simply did not include them in this analysis.

Figure 1:

23. This figure needs a lot of improvement. The font on this figure is rather small and should be improved. The scale bar is barely visible. The icons in the legend should be lined up nicely. It would also be helpful to have the site names included.

- Corrected in manuscript

Figure 3 and 7:

24. Why are these two figures separated since figure 3 also has mobile taxa? They are both very nice but could be combined.

- The two figures are now combined

Supplementary material:

Figure 1 and 2:

25. Please check the site names for this figure. In 1a and 2a the station names are included, but in 1b and 2b the station numbers are included instead. Be consistent and just use site names or station number throughout the entire text. Additionally, the colors corresponding the station numbers in Figure 1 also seem to be flipped for Sill (station 13 according to table 1) and WTR (station 4 according to table 1).

- Corrected in manuscript

Reviewer #2: This work is worthy of publication – it focuses on important benthic habitats that provide Essential Fish Habitat and support keystone species. The study is focused on a high latitude ecosystem that is susceptible to climate change, and it is valuable to understand current species’ distributions from which to measure future change. However, I have a number of comments and concerns that I think need to be addressed before this is suitable for publication. I therefore recommend publication with major revisions.

Many thanks for your extremely helpful review of this manuscript. We have incorporated the changes, and hope this now addresses all concerns. 

6. PLOS authors have the option to publish the peer review history of their article (what does this mean?). If published, this will include your full peer review and any attached files.

Do you want your identity to be public for this peer review? For information about this choice, including consent withdrawal, please see our Privacy Policy.

Reviewer #1: Yes: Heidi Kristina Meyer

Reviewer #2: No

REVIEW

Deep benthic coral habitats of Glacier Bay National Park and Preserve, Alaska Hartill et al. 

General comments 

The manuscript is generally well-written, but sentences are sometimes unnecessarily cumbersome. I have suggested some edits that may help clarify and streamline the manuscript. 

The authors use a mix of deep-sea and cold-water corals in describing their focal community. I recommend picking a term and using it consistently. Since the focus of the paper is a high latitude, relatively shallow ecosystem, I would use CWC instead of deep-sea. 

- Corrected in manuscript

This work is worthy of publication – it focuses on important benthic habitats that provide Essential Fish Habitat and support keystone species. The study is focused on a high latitude ecosystem that is susceptible to climate change, and it is valuable to understand current species’ distributions from which to measure future change. However, I have a number of comments and concerns that I think need to be addressed before this is suitable for publication. I therefore recommend publication with major revisions. 

Title 

Benthic doesn’t seem necessary.

- Corrected in manuscript

Abstract 

L1: Suggest changing to ‘...southeastern Alaska, comprising a system of fjords.... 

- Corrected in manuscript

Keywords 

I suggest adding cold-water corals instead of, or in addition to, diversity 

- Corrected in manuscript

Introduction 

L28: while this statement is still true, the authors should acknowledge that there have been many CWC studies conducted, particularly over the past decade. Freiwald and Roberts is quite out of date now. Amend statement and add new references. 

- Corrected in manuscript, lines 53-55

L30: Add comma after ‘foundation for,’ and ‘that sustain,’ 

- Corrected in manuscript

L33: Move ‘in areas such as ......Scandanavia’ to L43, replacing ‘at these four locations’ 

- Corrected in manuscript

L34: Add ‘than usual’ after ‘shallower depths’ 

- Corrected in manuscript

L39: Explain why a freshwater layer would result in reduced light. I also suggest replacing ‘enhanced darkening’ with ‘reduced light’ 

- Corrected in manuscript, lines 66-71

L47: Delete ‘Among these high latitude fjord systems where’ 

- Corrected in manuscript

L49: Suggest adding (commonly known as the Red Tree Coral) after Primnoa pacifica 

- Corrected in manuscript

L50: Replace ‘deep-water’ with ‘deeper’ – 20 m cannot be considered deep-water 

- Corrected in manuscript

L55: Replace ‘is’ with ‘encompasses’ after GBNPP 

- Corrected in manuscript

L65: Delete ‘scale’.

- Corrected in manuscript

L69: Delete ‘commercial fishing activity and other’ and add a phrase or sentence that acknowledges climate change as a major anthropological factor

- Corrected in manuscript

L74-84: This section describes the area in detail, but there is no frame of reference since these places are not marked on the map. The authors should add labels to Fig. 1, showing some of the more important features described in this section. The map also needs study site labels. 

- Corrected in manuscript

L88: I assume ‘vegetation’ is terrestrial not aquatic, but please clarify 

- Corrected in manuscript

L91-92: I don’t see how sedimentation creates stratification, which is the implication of this sentence. I suggest removing ‘due to increased sedimentation’ and starting the next sentence with. ‘Increased sedimentation from run-off leads to lower light...’ 

- corrected in manuscript, rephrased

L94: Replace ‘shallow ocean’ with ‘photic zone’

- corrected in manuscript

L95: Replace ‘related material’ with ‘other organic’ 

- corrected in manuscript

L99: Explain why optimal conditions occur ‘where fjord and shallow sill processes meet’ 

- corrected in manuscript, explained and expanded lines 185-222

L100: Do these statements apply to all/most fjords or are they specific to the study area? Clarify, and if the latter, provide a depth for the ‘deep central basin’ 

- corrected in manuscript

L109: Suggest deleting ‘dichotomously branching’, which is a specific taxonomic term and should be defined if used. 

- Corrected in manuscript

L112: replace ‘in’ with ‘of’ 

- Corrected in manuscript

L115-119: This is a long and rather cumbersome sentence. I suggest re-wording to ‘The large, complex structure of P. pacifica colonies provides habitat for a diverse community of associated species, some of which (such as rockfish and crabs) are economically important (Stone....etc). 

- Corrected in manuscript

L119: Suggest re-wording to ‘Primnoa pacifica therefore exhibits keystone.....’ 

- Corrected in manuscript, 264-266

L122: This is a rather vague and unsubstantiated statement. Explain or delete. 

- Corrected in manuscript

L126-128: The authors reported a study in 2010 that surveyed areas 180 m deep, so this survey is actually isn’t the first deep sea (defined by NOAA as >50 m) study. Suggest re-phrasing to ‘This study expands on earlier surveys in the GBNPP to assess benthic community structure from 100- 300 m using a remotely operated vehicle (ROV). Note – the 100-300 m depth range is stated by the authors, but table 1 indicates the shallowest depth surveyed was 197 m. Clarify/adjust 

- Corrected in manuscript, “table 1 indicates the shallowest depth surveyed was 197 m” – in table one, the column “transect start depth” was where the “197m” was reported, and “transect start depth” referred to the depth at which the transect began, and since the transects were vertical ascents, 197m would not be the shallowest depth surveyed. But I changed that column anyways to “length of transect” and upon reviewing my raw data for depths found that I had misreported the deepest depth as 300m when it was 420m 

Materials and methods 

This section could be better organized. Start with site selection, describe the vehicle and associated instruments in full detail, describe survey methods, then statistical analysis and data treatment. Specific comments below. 

- Corrected in manuscript

L136: Delete a priori – not appropriate context. L137-139: Move to later section on ROV surveys 

- Corrected in manuscript

L139-142: This section says sites were selected based on existing knowledge of coral distribution, but L136 states multibeam was used for site selection – clarify. 

- Corrected in manuscript, an existing knowledge of the substrate on which Primnoa grows helped researchers choose sites based on multibeam data of the basin, line 280

L148: Table 1. Site name abbreviations should be added to the map. Error (st. dev) should be included in the temp and salinity averages. Transect depth range should be used instead of start depth as transects move up steep walls. Delete Station # - it has no context. The total transect areas are very small for ROV surveys. Please explain why in the text. 

- Corrected in manuscript, it was a multipurpose expedition

L150: This sentence is rather confusing. Vertical transects imply ascending a wall (maybe the authors mean the camera was oriented vertically?), whereas ‘along the seafloor’ implies a more horizontal aspect. The phrase ‘beginning near the deep central axis then moving to the base of the wall’ again implies some horizontal survey prior to the vertical. If this was the case for every dive, there would likely have been a mix of habitat types surveyed, which would have confounded the community analysis. 

- Corrected in manuscript, “2.2 Survey Method” lines 363-485 in “Revised Article with Changes Highlighted” file

L156-157: Unclear what this means – how does splitting the video into quadrats compensate for different transect lengths? 

- Corrected in manuscript, line 490-494

L158: ‘Portions were selected based on presence of epibenthic fauna’ – this sounds like a confounding factor for a community analysis – please explain. 

- Corrected in manuscript, lines 493-494

L164: Replace ‘A CTD....’ with A Sea-Bird SBE-19 data logger (Sea-Bird Electronics Inc.) was mounted to the ROV and collected...... should this be depth not density? Add units. 

- Corrected in manuscript

L169: Rephrase to ‘assess the variation’ – confirm implies an a priori assumption of uniformity. 

- Corrected in manuscript, line 400

L175: Terminology – morphospecies isn’t used in the species richness table – or presumably the calculation. Do the authors mean species? Taxa? 

- Corrected in manuscript

L176: Total area surveyed – within a transect? Across transects? 5% seems rather arbitrary – how did the authors decide on this number? 

- Corrected in manuscript, 5% is based on recent work defining rare species at local and regional spatial scales as those that occur at this threshold. Mouillot et al. 2013 is now cited in the text.

L180: How were size classes estimated? The description says coral cover was assessed in 2 dimensions but it is unclear how this translated to size class. 

- Primnoa pacifica was measured in two ways – size classes were assessed separately from the 2d percent cover of the transect. Lines 660-687

L189: How was Primer E used to estimate diversity? 

- Corrected in manuscript “2.4 Statistical Analysis”

L191: which data? Percent cover is mentioned earlier, but if % cover was the only variable used, why/how was the data standardized? Explain why the square root transform was used 

- Corrected in manuscript, in “2.4 Statistical Analysis” but for more see Ahrens, Cox and Budhwar 1990 “Use of the arcsine and root transformations for subjectively determined percent data” 

L194: One way ANOSIM? Rephrase, ‘....were conducted to determine whether there were significant differences in community assemblages among sites..’ – with distance from glacier as a factor? 

- Corrected in manuscript

Results and Discussion 

L203: One would expect steep hard substrate on the walls, but not in the central axis. The authors were targeting the walls so its reasonable to expect this habitat would dominate their observations. Suggest deleting ‘...as would be expected....’ 

- Corrected in manuscript

L215: Incorrect use of morphospecies. Use different term – e.g. taxa. 

- Corrected in manuscript

L220: I suggest using scientific names rather than common names (ophiuroids, holothurians, echinoids, asteroids) 

- Corrected in manuscript

L228: Table 2 doesn’t present by site as the table title suggests - species lists are grouped by geographic location. My preference as a reader would be to have all sites listed separately, not combined. It would also be helpful to have the site abbreviations added to the location column titles. 

- Doing that would make the table very cumbersome … however I corrected it to say “site groupings” in title and added site abbreviations to columns

L231: Spell out Primnoa pacifica in the figure legend 

- Corrected in manuscript

L239: Clarify whether these species richness values are low/high in the context of other fjords etc (in which case, supply references), or whether the authors are simply using relative terms. If so, use relative terms to describe observations. 

- Corrected in manuscript, using relative terms

L242: Delete ‘The data at’. The difficulty is not using an ROV, but that the coral canopy hides the below-canopy substrate. Re-word. 

- Corrected in manuscript

L249-254: This section repeats L241-244. Combine and streamline information.

- Corrected in manuscript

L257: Either delete ‘....and species associated....’ Or expand. It’s currently rather a loose end. 

- Corrected in manuscript

L259-262: While this statement may be valid, it is not an example of depth-related differences – in fact as a general rule food decreases with depth. 

- Clarified in paragraph lines 948-961

L262-264: While this may be true, the context is unclear. There is no apparent depth related species turnover observed between the shallower and deeper sites in this study. 

- Corrected in manuscript lines 948-961

L264-265: Needs reference. Also “restricted’ is rather subjective and there are many deep sea species to which this statement does not apply – Pp, being one of them. Again, it’s also unclear what the context/argument is here. There doesn’t seem to be any evidence in this paper to support zonation or species turnover by depth. There are larger colonies of Pp in the west arm mid sites, but it is unclear why. There are too few replicates and too many other confounding factors to say with certainty. 

- Clarified in paragraph lines 948-961

L269: Expand on ‘patchiness of benthic community assemblages’ – are the differences associated with underlying geology? Some physical factor such as locally elevated currents? Nothing apparent? 

- Corrected in manuscript

L277: The Michaelis et al 2019 was in the North Sea – clarify the similarities between that project and the current study. 

- Corrected in manuscript, lines 1022-1027

L278: It is unclear which spatial gradients the authors are referring to. 

- Corrected in manuscript, deleted

L283-284: These sites have very different underlying geology – wouldn’t this be a primary reason for the differences? 

- Corrected in manuscript, line 1056-1057

L289-290: The text states that the taxa that contributed to 50% of dissimilarity between C1 and C2 were brachiopods, hydrozoan turf and barnacles. SI Table 1 shows similarity (not dissimilarity) within zones. The table only has species names not common names as in the text. I searched the paper and found the taxonomic name for barnacles and brachiopods but the reader should not have to rummage through the document to find pertinent information. SI Table 1 has a comparison of % similarity within zones, but the figures for barnacles, brachiopods and turf in the far zone columns do not add up to 50%. Something needs to be adjusted/clarified. C1 was also classified as primarily soft sediment, but Figure 4 shows C1 assemblages are dominated by hard substrate fauna. 

- Corrected in manuscript

L302-303. The C2 site was furthest from a glacier as stated in the text; however the distance between C1 and C2 was 1.5 km, so distance does not seem to explain the difference in clustering between these two sites. 

- Corrected in manuscript 

L305-307: The MDS plot does not seem to support the statement in the text that the MDS plot ‘suggests patterns of fidelity in species dominance determined not only by their proximity to glaciers but also by which fjords they inhabit’. Sites J1 J2, WTR and SILL are all near glacier sites but are in different clusters. HK, TB and C1 are similar distances from glaciers but do not cluster together. WTR, SILL (east) clusters closer with HK and TB (west) than WD and GP (east). It seems that substrate type may be driving the clustering of C1, WD and GP rather than proximity to glaciers or fjord arm. 

- Corrected in manuscript, used complete-linkage cluster dendrogram instead of group average, showing the placement of the sites in a more digestible order

L323-324: ‘...support our findings of lower diversity and abundance at the near glacier sites’. While this is true for J1 and J2, sites WTR and SILL have similar species richness to the mid sites HK and TB which are 40 km from the glaciers. 

- Corrected in manuscript

L363: It is unclear what the authors mean by ‘classical succession’. Retreating glaciers exposing hard substrate is not an example of ecological succession, which occurs when pioneering species modify the habitat to create conditions appropriate for the successive species. 

- Corrected in manuscript, rephrased.

Conclusions 

L412: This statement as written is not accurate – there have been other studies of Pp communities at depths up to 180m, which is defined as deep-sea.

- Corrected in manuscript

L415: Not consistently according to the data in this paper. 

- Corrected in manuscript, rephrased

Figures 

Fig 1: add site abbreviations and important features that will help interpret the data. Replace ‘studies’ with ‘study’ in caption. 

- Corrected in manuscript

Fig 3: Spell Primnoa pacifica the first instance it occurs in a caption, and P. pacifica thereafter 

- Corrected in manuscript

Fig 7: same comment as above 

- Figure 3 and 7 combined

SI Table 1. Add common names to species list to enable comparison with other tables in text. 

- Corrected in SI

Why is species contribution to dissimilarity between zones and similarity within? Is the table meant to be set up this way? If so, explain why in the table caption. The averages on the bottom row are clearly not averages of the columns. Explain why in the table caption. 

- Corrected in manuscript, all similarity now…

Fig 1 b: The sites are numbered, not named as in the other graphs. Fig 2 b: Same comment as above 

- Corrected in manuscript

---

## [Editor Report · Decision Letter 1]

17 Jul 2020

Deep coral habitats of Glacier Bay National Park and Preserve, Alaska

PONE-D-20-05571R1

Dear Dr. Hartill,

We’re pleased to inform you that your manuscript has been judged scientifically suitable for publication and will be formally accepted for publication once it meets all outstanding technical requirements.

Kind regards,

Andrew Davies

Academic Editor

PLOS ONE
---

## [Editor Report · Acceptance letter]

24 Jul 2020

PONE-D-20-05571R1 

Deep coral habitats of Glacier Bay National Park and Preserve, Alaska 

Dear Dr. Hartill:

I'm pleased to inform you that your manuscript has been deemed suitable for publication in PLOS ONE. Congratulations! Your manuscript is now with our production department. 

Kind regards, 

on behalf of

Dr Andrew Davies 

Academic Editor

PLOS ONE